# Combined Merkel Cell Carcinoma and Squamous Cell Carcinoma: A Systematic Review

**DOI:** 10.3390/cancers16020411

**Published:** 2024-01-18

**Authors:** Elisa Ríos-Viñuela, Fatima Mayo-Martínez, Eduardo Nagore, David Millan-Esteban, Celia Requena, Onofre Sanmartín, Beatriz Llombart

**Affiliations:** 1Escuela de Doctorado, Universidad Católica de Valencia San Vicente Mártir, 46001 Valencia, Spain; 2Department of Dermatology, Fundación Instituto Valenciano de Oncología, 46009 Valencia, Spainenagore@fivo.org (E.N.); osanmartinj@gmail.com (O.S.); 3School of Medicine, Universidad Católica de Valencia San Vicente Mártir, 46001 Valencia, Spain

**Keywords:** Merkel cell carcinoma, combined Merkel cell carcinoma, divergent Merkel cell carcinoma, combined squamous cell carcinoma and Merkel cell carcinoma, Merkel cell polyomavirus

## Abstract

**Simple Summary:**

Although considerable advances have been made in the understanding of the pathogenesis of Merkel cell carcinoma (MCC), many questions remain unanswered, particularly regarding the clinical characteristics and behavior of the less frequent, combined MCC and squamous cell carcinoma (SCC) tumors, as well as the origin of each element of these combined tumors. The main objective of this systematic review was to assess whether combined MCC and SCC tumors are associated with a worse prognosis than pure MCC. The results of this study showed similarly aggressive behavior and equivalent prognosis for combined MCC and SCC and pure, Merkel cell polyomavirus (MCPyV)-negative MCC. Moreover, preliminary data strongly suggest that all MCPyV-negative MCC tumors, whether combined or pure, are part of a common spectrum.

**Abstract:**

Combined Merkel cell carcinoma (MCC) and squamous cell carcinoma (SCC) have classically been regarded as more aggressive than conventional, pure, Merkel cell polyomavirus (MCPyV)-positive MCC. It is still unknown whether combined MCC and SCC are more aggressive than pure, MCPyV-negative MCC, and the origin of both the SCC and MCC elements of these combined tumors has not been elucidated. The main objective of this systematic review was to assess whether combined MCC and SCC tumors are associated with a worse prognosis than pure MCC; the secondary goals were the characterization of the clinical and histopathological features of these combined neoplasms. A total of 38 studies, including 152 patients, were selected for review. In total, 76% of the cases were MCPyV-negative, whereas 4% were MCPyV-positive. The most frequent histopathological pattern was that of an SCC in situ combined with a dermal MCC (36%), followed by both an in situ and invasive SCC combined with a dermal MCC (20%). Forty-seven percent of all cases fitted in the morphology of the so-called “collision tumors”. Three combined MCC cases that would fit in the morphological category of collision tumors presented both squamous and neuroendocrine elements in their respective nodal metastases. The mean overall survival was 36 months, comparable to that of pure, MCPyV-negative MCC. This review found similarly aggressive behavior for combined MCC and SCC and pure, MCPyV-negative MCC. Preliminary data strongly suggest that all MCPyV-negative MCC tumors, whether combined or pure, are part of a common spectrum.

## 1. Introduction

Since its first description by Toker in 1972, considerable advances have been made in the understanding of the pathogenesis of Merkel cell carcinoma (MCC) [1,2,3]. In 2008, Feng et al. demonstrated the integration of a previously unknown polyomavirus (MCPyV) into the genome of the MCC tumor cells [4]. Further studies confirmed MCPyV integration in about 80% of MCC cases and demonstrated the crucial role of both large T and small T viral antigens in transformation and neoplastic proliferation [3,5,6]. The large T antigen can be detected by immunohistochemistry with the CM2B4 antibody and is one of the currently preferred methods of detecting the virus’ presence. On the other hand, MCPyV-negative MCC cases (much more represented in countries with high UV exposure) have been shown to have a high mutational burden, similar to that of UV-induced keratinocytic tumors, and are thought to be driven by DNA damage caused by chronic UV damage [3,7,8]. Moreover, between 10 and 38% MCPyV-negative MCCs can be detected in close proximity to other UV-driven epithelial tumors and, although in fewer instances, divergent differentiation to squamous cell carcinoma has also been described, almost invariably in MCPyV-negative MCC [2,3,9]. All of these combined MCC and SCC tumors have been described in the literature with variable terminology, and they have further contributed to the growing body of evidence pointing to the two distinct origins of MCC that converge in a common neuroendocrine phenotype: (a) one driven by the integration of the MCPyV genome within the tumoral cells, of probable dermal origin, and characterized by a lower mutation burden, and (b) another one driven by UV radiation, of epidermal keratinocytic origin, characterized by being MCPyV-negative, and a high mutational burden [2,10,11,12]. Furthermore, several studies have associated MCPyV-negative tumors with a poorer prognosis, being more likely to present with advanced disease, and displaying a higher risk of progression [7,9,13,14]. In spite of significant progress in the understanding of MCPyV-negative MCC, its low incidence and the heterogeneity of the body of literature reporting combined MCC and SCC cause many questions to remain unanswered, particularly whether combined MCC and SCC tumors are associated with a poorer prognosis compared to both pure MCPyV+ and MCPyV-MCC, and whether combined or collision MCC and SCC tumors (even those without clear divergent differentiation) indeed derive from a common precursor. The aim of this systematic review was to assess whether combined MCC and SCC tumors are associated with a worse prognosis than pure MCC. The secondary outcomes were the characterization of the clinical and histopathological features of combined MCC and SCC tumors, as well as their management strategies. 

## 2. Materials and Methods

### 2.1. Study Design

The design of this systematic review was performed according to the Reporting Items for Systematic Reviews and Meta-analysis (PRISMA) guidelines [15,16] (Appendix A). The protocol was registered on the International prospective register of systematic reviews (PROSPERO) before searches, data extraction, and data analysis (CRD42023447096).

### 2.2. Eligibility Criteria

#### Inclusion and Exclusion Criteria

Given the paucity of available data in the literature, little restrictions regarding the type of article were imposed. All study types were included, whether randomized or non-randomized. Given the scarcity of accessible data in the literature, single case reports were also finally included, despite their expected high positive publication bias. To be included, studies needed to include patients with combined MCC and SCC, confirmed histologically. Studies conducted in both men and women of all ages were included, without restriction by location. Studies needed to qualify following critical appraisal by three separate reviewers (ERV, FMM, and BL).

Studies reporting combined MCC and tumors other than SCC were excluded, as were studies reporting incomplete information. Conference proceedings, abstracts, and other unpublished studies were not retrieved. Narrative reviews, systematic reviews, and meta-analyses were also excluded. Finally, articles with incomplete information were not included for review either. Only articles written in English, Spanish, or French language were considered for this systematic review. 

### 2.3. Search Strategy

The primary search focused on studies reporting patients with combined MCC and SCC tumors, confirmed histologically, published between January 1972 and June 2023. Three independent reviewers (ERV, FMM, and BL) conducted the electronic search. No language restriction or date restriction was placed on this search. We applied forward and backward snowballing of the identified relevant papers and adapted the search in case of additional relevant studies. The PECO strategy was used to develop the search criteria for the electronic databases. The PECO consisted of terms for combined Merkel cell carcinoma and squamous cell carcinoma. The bibliographic records retrieved were imported and deduplicated using the software SR-Accelerator. Key search terms included “Merkel cell carcinoma”, “combined Merkel cell carcinoma”, “collision Merkel cell carcinoma” in combination with “squamous cell carcinoma”, “combined squamous cell carcinoma”, and “collision squamous cell carcinoma” for the OVID Medline and Cochrane databases. Search terms were restricted to “Combined Merkel cell carcinoma AND squamous cell carcinoma” for the OVID Embase and Web of Science databases. 

### 2.4. Data Extraction

Three independent reviewers (ERV, FMM, and BL) examined the titles and abstracts of all studies initially identified via the search strategy. Articles fulfilling the inclusion/exclusion criteria were selected, and full texts were retrieved. Two independent authors (ERV and BL) then independently checked the full texts, excluded articles that were (justifiably) not eligible and extracted the data. Finally, a consensus was reached with the collaboration of all authors. The collected data from the studies included the following: first author, year of publication, study design, sample size, global location, patient demographics (age, sex, precedent of sun exposure, and precedent of other skin tumors), clinical characteristics (breakdown of duration/size/location, presence of lymph node metastasis at diagnosis, and presence of visceral metastases at diagnosis), histological characteristics (MCPyV status (both immunohistochemistry and molecular biology), histological pattern, characteristics of the squamous component, Breslow, presence of ulceration, presence, and characteristics of inflammatory infiltrate, mitosis, lympho-vascular invasion, elastosis, and immunohistochemical studies), and treatment strategies (conventional or Mohs surgery, adjuvant or therapeutical radiotherapy, chemotherapy, and immunotherapy). Missing data were expected, and after careful consideration, the reviewers agreed that study investigators would not be contacted for any unreported data/additional details (the reviewers expected that missing data would be hard to retrieve since an important proportion of the cases was expected to be old, and much of the missing information was expected to not have been reported because it had never been studied in the first place). Following discussion among all the reviewers, it was decided that the level of evidence (LE) of each article would be determined based on the Oxford 2011 Levels of Evidence and included in the extraction data table.

### 2.5. Risk of Bias

Due to the rarity of the condition, high levels of bias were expected. Included studies may suffer from a positive reporting bias. There is also a risk of confounding incorrect conclusions regarding correlating factors. We aimed, however, to identify studies that met our inclusion criteria and which carry risks of confounding and bias that are not critical. Two researchers (ERV and BL) independently assessed the methodological quality of the selected studies using the Joanna Briggs Institute (JBI) Critical Appraisal Tools for case reports, case series, or quasi-experimental studies (according to the study type). The observational trials and the cohort studies were considered intervention studies; thus, the JBI critical appraisal tool for quasi-experimental studies was applied. In case of disagreement between the scores provided, a consensus was reached via the collaboration of all authors.

### 2.6. Statistical Analysis

A narrative synthesis and construction of descriptive summary tables were carried out for all the included studies. The aforementioned extracted data were analyzed in a descriptive manner. 

## 3. Results

### 3.1. Study Selection

A PRISMA flow diagram provides details of the literature search (Figure 1).

### 3.2. Study Quality and Bias Results

Because of the rarity of the condition, the overall quality of the included studies was low. Based on the Oxford 2011 Levels of Evidence guidelines, 18 studies were level 4, whereas 20 studies were level 5 (see Appendix A). The JBI appraisal tool scores of included studies are reported in Appendix A. Regarding the 18 case series studies, only 2 studies conducted the statistical analysis. 

### 3.3. Study and Population Characteristics

A descriptive statistical analysis of the main characteristics of the patients included in the selected studies is summarized in Table 1 (clinical characteristics and outcome) and Table 2 (histopathological characteristics). The complete clinical and histological characteristics of the patients included in each selected study can be found in Appendix A. Of the 38 included studies, 19 were case series and 19 were case reports. No experimental studies could be retrieved. The 38 selected studies included a total of 152 patients with combined MCC and SCC. The mean age of the cohort was 78 years (median 78). Fifty-nine percent of the patients were men, whereas forty-one percent were women. The mean tumor size was 3 cm (median 2.5 cm), and the most frequent location was the head and neck area (56%). A history of immunosuppression or previous skin cancer, tumor stage at diagnosis, and initial management, which could be all considered relevant variables, were not reported in as many as 79% of all cases (see Table 1). Ninety-five percent of the cases that reported MCPyV status were MCPyV-negative, whereas five percent were MCPyV-positive. Of the cases that reported CK20 staining, 85% were CK20-positive, whereas 15% were CK20-negative, although information was not available in almost half of the patients (45%). The most frequent histopathological pattern was that of an SCC in situ combined with a dermal MCC (36%), followed by both an in situ and invasive SCC combined with a dermal MCC (20%) (Table 2). Forty-seven percent of all cases fitted in the morphology of the so-called “collision tumors”, whereas nine percent displayed areas of what has been described as “divergent differentiation”. Eight percent of all cases showed areas consistent with both patterns. Forty-four percent of cases developed at least nodal metastases throughout the course of the disease. Information regarding subsequent treatment was not provided in the vast majority of the selected studies. The mean overall survival was 36 months (the median was 41), whereas the mean follow-up was 24 months (the median was 23). The reviewers were unable to retrieve any more specific information on disease-free survival and progression-free survival, as detailed information regarding survival and follow-up was unavailable for a majority of the cases. 

## 4. Discussion

Overall, there was a lack of high-quality evidence reporting on the behavior and characteristics of combined SCC and MCC. The extensive search strategy carried out in this review could not retrieve any experimental studies; all the available evidence regarding combined SCC and MCC tumors comes from descriptive case series and reports, with barely any analytic studies that compared combined and pure MCC tumors. The following systematic review aims to compile the best available evidence on combined SCC and MCC, focusing on its behavior and prognosis, as well as its clinical and histopathological characteristics and management strategies. Because of the heterogeneity of the selected studies and the differences in data reporting, it was not possible to perform a meta-analysis.

Ever since the description of the MCPyV and its role in MCC tumorigenesis, there has been a growing body of research devoted to untangling the enigma of MCC’s cell of origin, and the last decade has witnessed great advances in this field, particularly with the identification of a keratinocytic origin for MCPyV-negative MCC [3,11,17]. Interestingly, despite a seemingly unrelated origin to Merkel cells, the preservation of the mechanosensory ability has recently been demonstrated in MCC [18]. There has also been a growing interest in the study of combined SCC and MCC and the relationship between the squamous and neuroendocrine components of these collision and/or divergent tumors [8,19,20]. However, most of these studies focus on histopathological and molecular analysis and provide little information on relevant clinical details, such as patients’ clinical background, tumor staging, treatment strategies, and disease behavior and prognosis [3,8,10,21,22,23,24,25]. Judging from the available data in the few reported case series, it is generally accepted that combined SCC and MCC tumors have a worse prognosis than pure, MCPyV-positive MCC, but this has never been studied comprehensively [2,9,26,27]. Only one study compared a cohort of 26 combined SCC and MCC tumors with 20 cases of pure MCC (although they did not specify whether these pure tumors were MCPyV-positive or -negative) and found that patients with combined tumors had a higher metastatic rate (77% vs. 40%) and lower survival (41 vs. 54 months) [28]. Furthermore, there is virtually no information comparing the prognosis of combined SCC and MCC to pure, MCPyV-negative MCC, which is already believed to behave more aggressively than its MCPyV-positive counterpart. Via our search, we could not retrieve any analytical study seeking to determine whether tumor behavior and patients’ survival varied from combined SCC and MCC tumors to forms of pure MCC. 

The aim of this systematic review was to assess whether combined MCC and SCC tumors are associated with a worse prognosis than pure MCC. However, the vast majority of the retrieved studies consisted of case series and case reports without a control or comparison group. Moreover, of the total of 152 patients included in the 38 selected studies, information on overall survival was unavailable for 67 of them (44%). The mean overall survival for the remaining cases was 36 months (median 41). Since the studies included did not compare combined and pure MCC cases for behavior, we sought to compare the information retrieved from the cohort to previously reported data on pure (MCPyV-positive and -negative) MCC. In a large cohort of 282 patients, Moshiri et al. found that MCPyV-negative MCC cases had a significantly increased risk of disease progression and death from the disease [7]. The median overall survival for MCPyV-negative MCC was 3.3 years against that of 4.6 years for MCPyV-positive MCC. A few years later, Harms et al. [27] and Naseri et al. [14] also found that the presence of MCPyV was associated with longer overall and disease-free survival. In the cohort retrieved from our review, the median overall survival of 41 months (3.4 years) is comparable to that of Moshiri’s study, hence suggesting that while combined SCC and MCC are probably associated with a more aggressive disease course than pure, MCPyV-positive MCC, its behavior seems to be similar to that of MCPyV-negative tumors. 

Although combined SCC and MCC are considered to be relatively rare [2,9,29,30], the association of in situ (and/or invasive) SCC and MCC in close proximity has been observed in as many as 38% of cases in some MCPyV-negative MCC series [3,29,31,32]. These tumors have generally been depicted as collision tumors, and it remains a matter of debate whether one component derives from the other or they rather represent a casual association of two tumors that share UV radiation as a common etiological agent (Figure 2) [2,3]. The vast majority of collision tumors reported to date were MCPyV-negative [2,3,19,30], with only a few exceptional MCPyV-positive cases [24,30,33]. On the other hand, a few case series and reports have studied the much rarer divergent MCC, also known as MCC, with aberrant differentiation [9,30,34,35,36,37]. These tumors display areas of abrupt transition to SCC within the MCC tumor, and both elements appear morphologically intermingled, with the merging of the neuroendocrine cells into squamous differentiation (Figure 3 and Figure 4). Although even more infrequent, cases of MCC with divergent differentiation to other lineages (sarcomatoid, eccrine, etc.) have also been reported [9]. Again, most of them are MCPyV-negative [2,9,19,30,35,36]. MCC with divergent differentiation has consensually been considered a single tumor with aberrant transformation into other lineages [2,9]. To support this hypothesis, there have been a few reports of MCC with divergent differentiation showing both neuroendocrine and squamous components, both in the primary tumor and in nodal metastases [9,30,35,36,37]. It should be noted that the nomenclature used to describe these two morphologically distinct combined tumors has been confusing and inconsistent, as some authors restricted the term “combined MCC” for the description of MCC with divergent differentiation, while others utilized it for both divergent and collision tumors. In more recent studies, authors have shifted to a simpler morphological classification based on the microanatomical location of the neuroendocrine and the squamous elements of the combined tumor (Figure 5) [10,23].

In the present cohort of 152 patients, the most frequent histopathological pattern was that of in situ (+/− invasive) SCC combined with dermal MCC (56%). Forty-seven percent of all cases fitted in the morphology of the so-called collision tumors, whereas nine percent displayed areas of what has been described as divergent differentiation. The present review found three combined MCC cases that would fit in the morphological category of collision tumors but still presented both squamous and neuroendocrine elements in their respective nodal metastases [30,36,37]. The presence of both elements in the metastatic tumor argues against the coincidental association of the SCC and MCC elements, even if they simulated a collision neoplasm in the primary tumor. A recent study by Kervarrec et al. sought to elucidate the controversial relationship between the two components of combined collision MCC and SCC tumors [3]. After individual dissection and whole exome sequencing of 4 combined tumors consisting of SCC in situ and MCPyV-negative MCC, they found that a significant percentage of somatic mutations were shared between SCC and MCC, with higher allelic frequents of the shared variants present in the MCC part. They also found a high mutational burden and a prominent UV signature. These results provided strong evidence that these MCPyV-negative MCC cases likely arose from the accompanying SCC in situ. A similar recent study by Harms et al. evaluating seven pairs of combined MCC and SCC in situ also found a highly significant mutational overlap between both squamous and neuroendocrine elements, including shared TP53 and/or RB1 mutations [23]. Both studies associated the shift to neuroendocrine phenotype with loss of Rb protein expression and increased SOX2 expression, probably via epigenetic changes [3,23]. Although still preliminary, all these recent developments point to the fact that regardless of a divergent or collision distribution, combined MCC and SCC tumors are merely two morphological and phenotypical variants of the same tumor that arise from a common keratinocytic precursor. Moreover, studies comparing both combined and pure MCPyV-negative tumors found no significant differences between the two, as they both share a high mutational burden, UV signature mutations, and inactivated mutations and/or decreased copy numbers in the *RB1* and *TP53* genes [38]. None of these findings is associated with MCPyV-positive MCC. Hence, it seems that all these morphologically diverse, MCPyV-negative MCC tumors, whether pure or combined, are all part of a common spectrum, encompassing all the possible presentations of UV-induced, MCPyV-negative MCC. It thus seems only fitting that, as this review has found, both combined and pure MCPyV-negative MCC share a similar, more aggressive behavior. 

## 5. Conclusions

To the best of our knowledge, this is the first systematic review seeking to elucidate the relationship between combined MCC and SCC and its pure, MCPyV-negative counterpart. The strength of this study resides in its extensive search strategy, including all available case series and reports that provided at least basic clinical information, with a relatively large sample size. However, this study had clear limitations. We found great heterogeneity within the selected studies: many were molecular or pathological studies that provided little clinical information, and the nomenclature used to refer to collision and divergent tumors was inconsistent and confusing. The use of immunohistochemical stains and procedures to evaluate for the presence of MCPyV were also greatly heterogeneous amongst the selected studies. This reveals the need for homogenous data reporting methods. Moreover, our search could only retrieve case series and reports, and thus, the overall quality of the included studies was low. Publication bias and sample-sized bias are expected to be present. All the aforementioned difficulties made it challenging to perform a meta-analytical study. 

In summary, the current systematic review revealed a similarly aggressive behavior and equivalent prognosis for combined MCC and SCC and pure, MCPyV-negative MCC. Moreover, preliminary data strongly suggest that all MCPyV-negative MCC tumors, whether combined or pure, are part of a common spectrum. Regardless of the previously utilized nomenclature, all cases of combined MCC and SCC, whether morphologically divergent or collision tumors, seem to represent a single tumoral entity, with biphasic differentiation to two cellular lineages. To improve the quality of evidence on this rare neoplasm, international collaboration is essential to standardize reporting methods and collect a larger number of patients to consequently obtain high-quality evidence to validate these preliminary data.

## Figures and Tables

**Figure 1 cancers-16-00411-f001:**
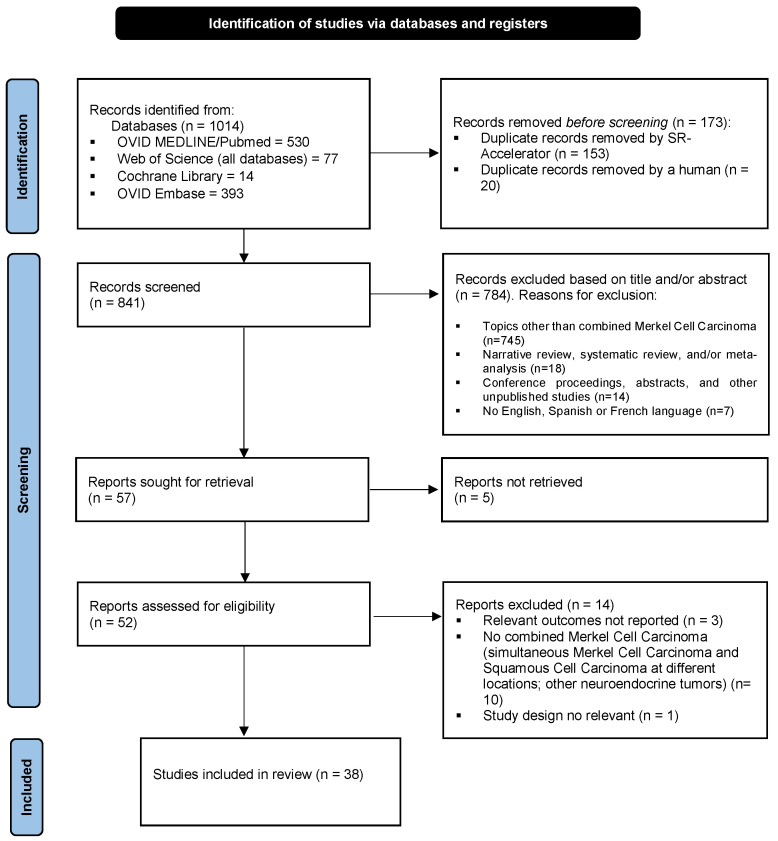
The flow chart with the different phases of the systematic literature search and selection of studies included in this review. The total number of search results was 1014 records. After elimination of duplicates (173), another 784 studies were excluded based on title and abstract screening, and finally, 14 studies were excluded after full-text assessment. Thus, the study selection resulted in a total of 38 relevant articles included in the present review of combined MCC and SCC (Appendix A).

**Figure 2 cancers-16-00411-f002:**
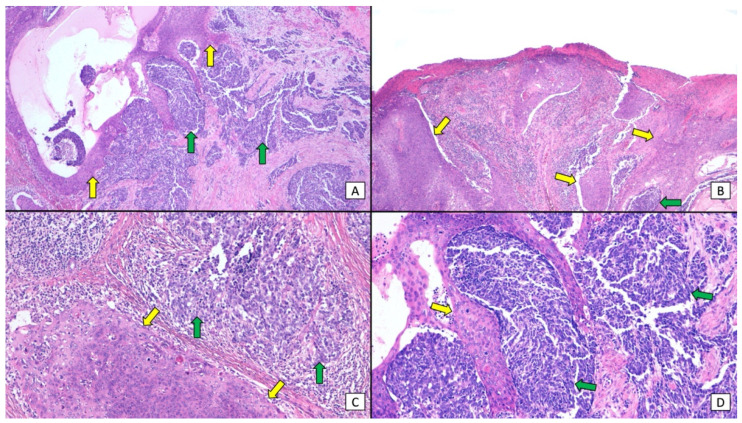
Combined MCC and SCC are classically depicted as a collision tumor. (**A**,**B**) (×200, original magnification) the neuroendocrine (green arrows) and squamous (yellow arrows) components of the tumor can be seen closely together, with sheets of small, basophilic tumoral cells intermingled with nests of atypical, squamous cells; (**C**,**D**) (×400, original magnification). A closer view highlights the dermal invasion of sheets of small, monomorphic, basophilic, tumoral cells (green arrows), as well as large nests of atypical squamous cells (yellow arrows). Both elements appear closely together, without areas of clear transition from one component to the other.

**Figure 3 cancers-16-00411-f003:**
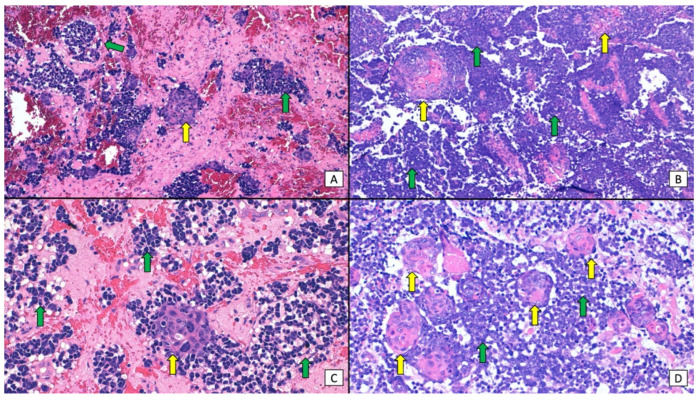
Combined MCC and SCC typically described as MCC with divergent differentiation. (**A**,**B**) (×200, original magnification) diffuse dermal invasion by sheets of small, basophilic, neuroendocrine cells (green arrows), with areas of abrupt transition into nests of atypical, squamous cells (yellow arrows); (**C**,**D**) (×400, original magnification). A closer view highlights the abrupt transition of the neuroendocrine cells (green arrows) into the squamous component (yellow arrows).

**Figure 4 cancers-16-00411-f004:**
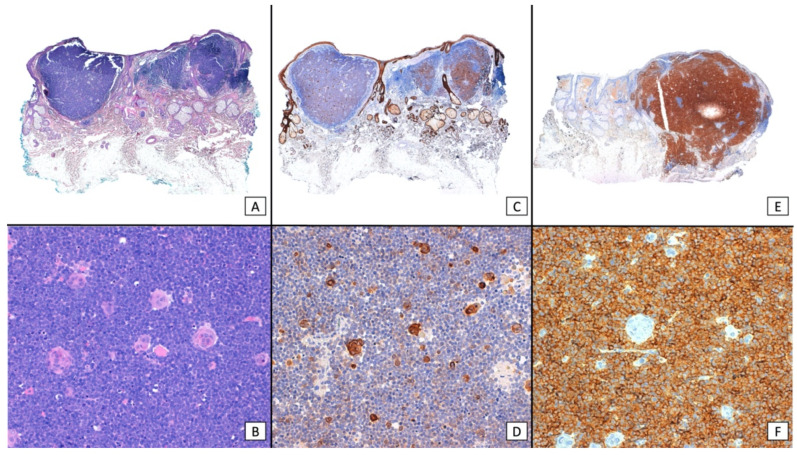
Immunohistochemical panel of combined MCC and SCC: (**A**) (×40, original magnification) shows a panoramic view of a dense, basophilic, dermal tumor; (**B**) (×200, original magnification) shows a closer view and highlights areas of abrupt transition into nests of squamous cells; (**C**) (×40, original magnification) the squamous nests show positive staining with CKAE1/AE3, whereas the basophilic, neuroendocrine cells remain negative; (**D**) (×200, original magnification) a closer view of the positive squamous nests; (**E**) (×40, original magnification) the tumor shows diffuse positive staining with synaptophysin; (**F**) (×200, original magnification) a closer view highlights intense positive staining for the neuroendocrine cells, whereas the squamous nests remain negative.

**Figure 5 cancers-16-00411-f005:**
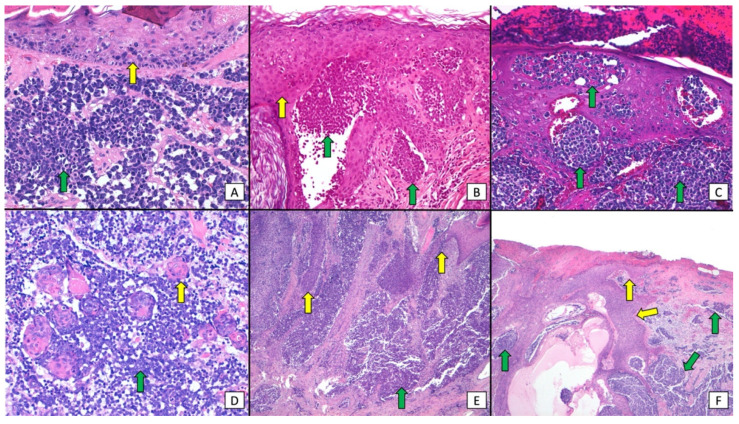
Morphological classification of combined MCC (green arrows) and SCC (yellow arrows) tumors. (**A**) (×400 original magnification) dermal MCC associated with intraepidermal SCC. (**B**) (×400 original magnification) Intraepidermal MCC associated with intraepidermal SCC. (**C**) (×400 original magnification) Intraepidermal and dermal MCC. (**D**) (×400 original magnification) Dermal MCC and SCC. (**E**) (×200 original magnification) Dermal MCC associated with intraepidermal and dermal SCC. (**F**) (×400 original magnification) Intraepidermal and dermal MCC associated with intraepidermal and dermal SCC.

**Table 1 cancers-16-00411-t001:** Main clinical characteristics of the systematic review cohort. Non-melanoma skin cancer (NMSC), Malignant melanoma (MM), Not available (NA). The mean age of the cohort was 78 years (median 78). The mean tumor size was 3 cm (median 2.5 cm). Mean overall survival was 36 months (median 41), whereas mean follow-up was 24 months (median 23).

Main Clinical Characteristics							
Sex							
Male				Female		NA	Total
89/152 (59%)				63/152 (41%)		0/152	152
Primary Tumor Location							
Head and Neck			Extremities		Trunk	NA	Total
85/152 (56%)			32/152 (21%)		20/152 (13%)	15/152 (10%)	152
Previous Skin Cancer							
NMSC	MM		NMSC + MM		None	NA	Total
37/152 (24%)	1/152 (<1%)		3/152 (2%)		5/152(3%)	106/152 (70%)	152
Immunosuppression							
Yes				No		NA	Total
20/152 (13%)				21/152 (14%)		111/152 (73%)	152
Tumor Stage at Diagnosis							
Local		Nodal		Nodal or Visceral		NA	Total
40/152 (26%)		14/152 (9%)		14/152 (9%)		84/152 (55%)	152
Progression to Metastatic Disease (at Least Nodal)							
Yes				No		NA	Total
67/152 (44%)				27/152 (18%)		58/152 (38%)	152

**Table 2 cancers-16-00411-t002:** Main histopathological characteristics of the systematic review cohort.

Main Histopathological Characteristics							
Histopathological Anatomical Distribution of SCC and MCC							
SCC Epidermis/MCC Dermis	SCC Epidermis + Dermis/MCC Dermis	SCC Epidermis/MCC Epidermis	SCC Epidermis/MCC Epidermis + Dermis	SCC Epidermis + Dermis/MCC Epidermis + Dermis	SCC Dermis/MCC Dermis	NA	Total
54/152 (36%)	31/152 (21%)	4/152 (3%)	3/152(2%)	2/152(1%)	16/152 (11%)	42/152 (28%)	152
Histopathological distribution of SCC and MCC as independent (collision) tumors or tumors with divergent differentiation							
Independent (collision) tumors		Tumors with divergent differentiation		Tumors showing both patterns		NA	Total
71/152 (47%)		14/152 (9%)		12/152 (8%)		55/152 (36%)	152
MCPyV status							
Negative IHC (CM2B4 large T antigen)	Negative PCR	Negative IHC and PCR	Positive IHC (CM2B4 large T antigen)	Positive PCR	Positive IHC and PCR	NA	Total
56/152 (37%)	6/152 (4%)	54/152 (36%)	3/152 (2%)	1/152 (<1%)	2/152 (1%)	30/152 (20%)	152
CK20 (IHC)							
CK20+			CK20−			NA	Total
71/152 (47%)			13/152 (9%)			68/152 (45%)	152

## Data Availability

Most of the data that support the findings of this study are available in the Appendix A. Further information will be provided by the corresponding authors, E.R.V. and B.L., upon reasonable request.

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
