# Peer review of "Combined Merkel Cell Carcinoma and Squamous Cell Carcinoma: A Systematic Review"

_cancers, 2024, doi:10.3390/cancers16020411_

Round 1
Reviewer 1 Report
Comments and Suggestions for Authors
This review manuscript described 38 studies including 152 cases from January 1972 to June 2023 to analyze Merkel cell carcinoma (MCC) and squamous cell carcinoma (SCC) combined tumors aiming to compare with pure MCC and to predict a prognosis. The analysis concludes with a similar aggressive behavior and equivalent prognosis for combined MCC/SCC and pure MCC. There are only a few minor things that need to be addressed.
1. Is the statement in the legend for Fig. 1 that "a total of 35" is accurate?
2. Table 2, "CM2B4" is not explained to general readers. It would be better to briefly mention "large tumor (LT) antigen-antibody" in the material and methods or mention "large T antigen" in the table.
3. Fig. 2 and Fig. 3 do not appear to be using x100 magnification (different magnification). please check again. In each figure (Fig. 2, 3, 4), the cells/tumors explained in the legend should be accurately indicated by using arrows in the figure for easier understanding.
Author Response
Thank you very much for taking the time to review this manuscript. Please find the detailed responses below and the corresponding revisions/corrections highlighted/in track changes in the re-submitted files. 1. Thank you very much for bringing this to our attention. The typing error has been corrected and the statement has been modified to 38 articles, as it is clearly stated throughout the rest of the manuscript and in the figure itself, as well as the supplementary material. 2. Thank you for bringing this to our attention. We have modified the table as requested, and in the introduction, while discussing MCC's pathogenesis we have also specified that the large viral T antigen can be detected by immunohistochemistry with the CM2B4 antibody. We haven't added this to the methods section because this section mainly related to the search strategy, and thus we thought it would be more appropriate elsewhere. 3. Thank you for bringing this to our attention. Figure 2A+2B are indeed 200 magnification, whereas 2C+2D are x400 magnification. Figure 3C+D are x400 magnification, whereas Figure 3A+B is 200 magnification. This has been corrected. We have also added arrows for easier understanding, as requested.Reviewer 2 Report
Comments and Suggestions for Authors
The meta-analysis by Ríos-Viñuela, E. et al. about combined squamous and Merkel carcinoma appears scientifically correct and well organized. It is difficult to point any significant flaw and is a really interesting review to my eyes.
I have a pair of major comments to make, only for your consideration:
- I understand that methods can be quite rigid, but you may contact the different authors asking for the missing information. This way, you might solve (well, try to solve, actually) the main limitation in the study, which is the lack of relevant data regarding MCPyV status (and other clinical and histological data) in a really significant number of cases. I know it is not realistic to expect great improvement in the data this way, but it seems “lazy” when I see written that you don´t have intention to contact other investigators requiring more data… (lines 129-130). May be you can remove this part of the sentence…
- The other issue is that in the text is mentioned (for example in the first –and only- paragraph of introduction and the second of discussion) either an epidermal keratinocytic cell or a dermal MCPyV “infected” cell as origin of MCC. Merkel cells (MCs) are still elusive and keratinocites / epidermal stem cells has been suggested as precursors (Becker et al., Nat Rev Dis Primers, 2017, 10, 399-408) as well as dermal fibroblasts (Sunshine et al., Oncogene, 2018, 37, 1409-16). However, MCs are very specialized cells despite their origin, and recently the preservation of the mechanosensory ability in MCC was demonstrated by the expression of Piezo2 (García-Mesa et al., J Pers Med, 2022, 12:894). It´s true that this reference corresponds to a short case series, with no reference to MCPyV status, but it should be considered when you discuss the origin of the MCC cell, because, as I mentioned at the beginning of this paragraph, the general impression after reading the whole text is that MCC have no relation with MCs.
There are also some minor comments:
- Abstract. Lines 26. I miss a word after “these combined”… May be neoplasms?
- Materials, section 2.3, line 106. Records were imported.
- Materials, section 2.4, line 128. Data was / were expected, depending if you expected a single or multiple data to be missed. Employ past tense in the sentence (were not contacted…).
- Materials, section 2.5, line 134. Were expected. Line 136; that met. Same as the previous issue
Comments on the Quality of English LanguageThe language is fine and I can only find minor issues.
Author Response
Thank you very much for taking the time to review this manuscript. Please find the detailed responses below and the corresponding revisions/corrections highlighted/in track changes in the re-submitted files. 1. We understand the reviewer's concern. The reason that we specifically make this statement is that it is supposed to appear in a through systematic review protocol, investigators are supposed to clearly state whether they will seek (or not) missing information. We carefully considered whether reaching out to authors of previous articles, but finally decided not to do so because a)some of the cases included are quite old and we think it unlikely to be able to retrieve these patient's records b) we believe that some of the missing relevant data (such as MCPyV status) are likely not reported because they were never performed, and we think it unlikely additional tests will be run in these cases just because ask the investigators c) we have previously tried to contact some investigators, for previous work, and it proved time-consuming, and little useful when it came to retrieving additional data that is not already reported in the published manuscripts. We can remove the sentence if the reviewer thinks it's best (we have left it as it is, as of now), but we believe it is honest to state it as it is, and we believe it is not unusual for it to appear in a systematic review protocol. 2. Thank you for bringing this to our attention. We had not previously mentioned it because of the lack of reference to MCPyV status, but we agree with the reviewer that it is worth mentioning. It has been added to the discussion section. 3. Thank you for bringing all these errors to our attention. The protocol was developed in advance of the search, and was uploaded to the PROSPERO site. The sentences were not in the correct tense because we had not yet changed it to the past tense once the review was conducted.Reviewer 3 Report
Comments and Suggestions for Authors
I found the review question of the authors very interesting and I do not have any major issues. Some points to improve the work
- it could be useful and transparent to have in the supplementary material the full search strategies for all the screened databases
- in the results section, you do not say so much on the outcome in terms of OS, DFS and so on, you only report mean and median survival. I think that starting with your spread sheet with all the data, you can perform something like a individual patient based meta-analysis considering each case report as a individual patient and of course the patients of the case series. For you convenience, here some papers on another topic (diagnosis and outcome of transmitted cancer in transplant recipients) which performed the same approach (doi 10.1111/ajt.12430 and doi 10.1007/s40620-020-00775-4).
Author Response
Thank you very much for taking the time to review this manuscript. Please find the detailed responses below and the corresponding revisions/corrections highlighted/in track changes in the re-submitted files. 1. Thank you for your comment. The search strategy is summarized in the PRISMA Flow chart. Moreover, the search protocol was uploaded to the Prospero site before the study was conducted (CRD42023447096). The search strategy has also been described in detail in the methods section, including the search terms used for each of the platforms, as well as the few modifications to the original protocol (such as the inclusion of single case reports) and their justification. We believe the search strategy is quite transparent as it is, as we have really strived to adhere to the PRISMA guidelines and provide every bit of information for our protocol. The only thing we could add as supplementary material are screenshots from the SR accelerator deduplication and selection process, unfortunately this is not possible because the program does not keep save these details for long once the process has been concluded. 2. Again, thank you for your valuable input. We have looked at the suggested articles and their strategies with great interest. The reason why we only report OS (and not DFS) is because detailed information about survival was lacking for many of the cases (as you can see in the spreadsheet, many articles do not even report OS). Initially, we tried to retrieve data regarding DFS and OS, but the follow-up for many cases was short (or non-existent altogether), and only very few reports specified DFS and OS. That's why we only chose OS, being the only parameter that was available for a majority of the studies. And since even this very basic parameter is missing in an important number of the reports, we thought it difficult to perform any further analysis.